# Joint Optimization of Transmit Waveform and Receive Filter with Pulse-to-Pulse Waveform Variations for MIMO GMTI

**DOI:** 10.3390/s19245575

**Published:** 2019-12-17

**Authors:** Zhoudan Lv, Feng He, Zaoyu Sun, Zhen Dong

**Affiliations:** College of Electronic Science and Technology, National University of Defense Technology, Changsha 410073, China; zhoudan_lv@163.com (Z.L.); sun_zyu@163.com (Z.S.); dongzhen@vip.sina.com (Z.D.)

**Keywords:** MIMO radar, GMTI, maximum SINR criterion, transmit waveform, receive filter optimize

## Abstract

Multi-input multi-output (MIMO) is usually defined as a radar system in which the transmit time and receive time, space and transform domain can be separated into multiple independent signals. Given the bandwidth and power constraints of the radar system, MIMO radar can improve its performance by optimize design transmit waveforms and receive filters, so as to achieve better performance in suppressing clutter and noise. In this paper, we cyclicly optimize the transmit waveform and receive filters, so as to maximize the output signal interference and noise ratio (SINR). From fixed pulse-to-pulse waveform to pulse-to-pulse waveform variations, we discuss the joint optimization under energy constraint, then extend it to optimizations under constant-envelope constraint and similarity constraint. Compared to optimization with fixed pulse-to-pulse waveform, the generalized optimization achieves higher output SINR and lower minimum detectable velocity (MDV), further improve the suppressing performance.

## 1. Introduction

Multi-input multi-output (MIMO) radar is a type of radar that uses multiple transmit antennas and multiple receive antennas. MIMO radar makes full use of the radar’s transmit freedom and can form a longer equivalent baseline [1,2], that means it have great potential in ground moving target indication (GMTI). MIMO system also provides more flexibility in the beampattern design, which makes the joint transceiver design of MIMO radar system to be possible [3,4,5]. MIMO radar also improves localization performance attainable thanks to the improved spatial diversity [6,7]. Its performance of detecting complex targets and invisible targets is obviously improved, the identifiability of the target parameters has also been significantly improved [8].

The waveform design of MIMO GMTI radar is divided into orthogonal waveform design [9,10,11] and non-orthogonal waveform design. MIMO GMTI radar can transmit orthogonal waveform and separate the transmit waveform with receive filters, so that to get longer virtual aperture and more data channels. Longer virtual aperture is beneficial in improving the performance of the MIMO GMTI radar and it also makes it possible for the radar to get more accurate parameter estimation and achieve smaller minimum detectable velocity (MDV) [12,13,14,15].

With the growing target detection requirements, as well as the continuous development of digital array technology and improvement of adaptive algorithm, cognitive-based transmitter-adaptive technology has become a hot topic in current MIMO radar research field. MIMO GMTI based on non-orthogonal waveform can improve moving target detection performance in certain tasks. Maximum output signal interference and noise ratio (SINR) criterion [16,17,18], mutual information (MI) criterion [19,20,21], minimum mean-square error (MSE) criterion [22] and ambiguity function [23,24] are usually used in waveform optimization. The adaptive waveform design under these criteria are discussed in Reference [25].

Recent years, with the swift development and wide application of MIMO radar, joint optimization of waveform and receive filters began to become a new research hot spot. The emergence of cognitive radar makes MIMO radar waveform design achieve great potential. It is also possible to optimize the transmit waveforms and receive filters together, so that to achieve better detection performance [26,27]. Waveform optimization based on Cram$\overset{´}{e}$r-Rao bounds (CRB) matrix is discussed in Reference [28], which demonstrates that minimize the traces of CBR matrix can improve the detection performance. The situation that extended targets exist and prior information is known is discussed in Reference [29], which uses cyclic iteration to jointly design transmit waveform and receive filters to achieve an optimized MIMO radar output SINR. Positive semi-definite relax algorithm is used in Reference [30] to transform the non-convex quadratic programming problem into convex problem, this provides an effective solution for MIMO GMTI radar waveform optimization.

Following the ideas above, the problem of cognitive transmit signal and receive filter design for a point-like target embedded in a high-reverberating environment is discussed in References [31,32]. The robust joint design of the transmit waveform and filter structure for polarimetric radar is discussed in Reference [33]. The joint design of transmit waveform and receive filters for MIMO radar STAP with fixed transmit waveform is discussed in Reference [2]. Reference [34] also discusses the transmit-receive filter design with covert communications focus. But for joint optimization with pulse-to-pulse waveform variations, these papers did not discuss it deeply.

In this paper, we use maximum output SINR criterion, generalize the transceiver optimization to MIMO radar system with pulse-to-pulse waveform variations. Compared to optimization with fixed pulse-to-pulse waveform, the output SINR is obviously increased. The output SINR is decided by transmit waveform and Space Time Adaptive Processing (STAP) optimized weight, we use cyclic iteration to jointly design transmit waveform and receive filters, so as to improve the MIMO GMTI performance.

The rest of paper is organized as follows. Section 2 establishes the signal model of target, clutter, jamming and noise. Section 3 is a review of transceiver joint optimization with fixed pulse-to-pulse waveform. Section 4 proposes the algorithms of transceiver joint optimization with pulse-to-pulse waveform variations. Section 5 provides numerical simulations to demonstrate the performance of the proposed algorithms. Finally, we draw the conclusion in Section 6.

*Notations*: Throughout this paper, matrices are denoted by bold capital letters and vectors are denoted by bold lowercase letters. Superscript ${\left(\cdot \right)}^{T}$ and ${\left(\cdot \right)}^{H}$ denote transpose and conjugate transpose, respectively. vec(·) denotes the operator of column-wise stacking a matrix and ⊗ represents Kronecker product. E[·] denotes the expectation of a random variable. tr(·) represents the trace of a square matrix. ${∥\cdot ∥}_{F}$ represents the Frobenius norm of a square matrix. A≻B(A⪰B) means A−B is positive definite (semi-definite).

## 2. Signal Model

Consider a MIMO radar with $N_{T}$ transmit antennas and $N_{R}$ receiver antennas, denote the waveform matrix of this system as S, where $S\in C^{N_{T}\times L}$, *L* is the code length.

### 2.1. Target

Let $S={\left[s_{1,m},s_{2,m},\cdots s_{N_{T},m}\right]}^{T}\in {}^{N_{T}\times L}$ denotes the waveform of the *m*th pulse, in which $s_{n,m}$ denotes the digital sampling of the *n*th transmitter in the *m*th pulse. To model the received signal of the targets, the return signal of a certain target direction at the *m*th pulse $t_{m}$ is written as:(1)$$T_{m}=\alpha_{t}e^{j2\pi \left(m-1\right)f_{d}T_{r}}a_{R}\left(\theta_{t}\right)a_{T}^{T}\left(\theta_{t}\right)S_{m}.$$

So the vectorized target return signal could be written as:(2)$$t_{m}=\mathrm{vec}\left(T_{m}\right)=\alpha_{t}e^{j2\pi \left(m-1\right)f_{d}T_{r}}\left(I_{L}⊗A\left(\theta_{t}\right)\right)\hat{s},$$
in which $\theta_{t}$ is the target direction of arrival (DOA), $A\left(\theta_{t}\right)=a_{R}\left(\theta_{t}\right)a_{T}^{T}\left(\theta_{t}\right)$, $\hat{s}=\mathrm{vec}\left(S\right)$, $a_{T}\left(\theta_{t}\right)\in C^{N_{T}\times 1}$ is the transmit array steering vector of $\theta_{t}$ and $a_{R}\left(\theta_{t}\right)\in C^{N_{T}\times 1}$ is the receive array steering vector, $\alpha_{t}$ denotes the target amplitude and $f_{d}$ denotes the target Doppler frequency. $I_{L}$ is a *L*- dimensional unit matrix.

### 2.2. Clutter

During the GMTI process, clutter suppression is a significant part. As the clutter distributes both in range and azimuth, it is generally defined as the superposition of all the scatters from all the distance units within the beam irradiation range. According to the range resolution of the radar system, the radar irradiation range is divided into multiple clutter rings, each of which is further divided into multiple independent clutter patches, all these clutter patches are regarded as scatters. The clutter patch division strategy is shown in Figure 1.

Here, we define a shift matrix: $P_{l}=P_{-l}^{T}\in C^{L\times L}$:(3)$$P_{l}\left(m,n\right)=\left\{\begin{array}{c} 1,\mathrm{if}\;m-n+l=0\\ 0,\mathrm{if}\;m-n+l\neq 0 \end{array}\right..$$

If the target is located in the *r*th range cell, the vectored return signal of the *k*th clutter patch in the (r+l) range cell is written as:(4)$$c_{l,k}=\alpha_{c,l,k}\left(I_{M}⊗I_{N_{R}}⊗P_{l}^{T}S^{T}\right)\left(u\left(f_{c,l,k}\right)⊗a_{R}\left(f_{c,l,k}\right)⊗a_{T}\left(f_{c,l,k}\right)\right),$$
where $u\left(f_{c,l,k}\right)={\left[1,\ldots ,e^{j2\pi \left(M-1\right)f_{c,l,k}T_{r}}\right]}^{T}$.

While computing the clutter covariance matrix, consider 2K+1 neighborhood range cells, each of which is divided into $N_{c}$ clutter patches, then the clutter is modeled as:(5)$$c=\sum_{l=-K}^{K}\sum_{k=1}^{N_{c}}c_{l,k}.$$

The clutter covariance matrix is written as:(6)$${\tilde{R}}_{c}=E\left[{\mathrm{cc}}_{}^{H}\right]=\sum_{l=-K}^{K}\sum_{k=1}^{N_{c}}\sigma_{c,l,k}^{2}{\tilde{S}}_{l}^{T}v_{c,l,k}v_{c,l,k}^{H}{\tilde{S}}_{l}^{*},$$
where ${\tilde{S}}_{l}=I_{M}⊗I_{N_{R}}⊗SP_{l}$, $v_{c,l,k}=u\left(f_{c,l,k}\right)⊗a_{R}\left(\theta_{c,l,k}\right)⊗a_{T}\left(\theta_{c,l,k}\right)$.

### 2.3. Jamming

For simplicity, in this paper, only take barrage noise jamming into consideration. The return signal of jamming is modeled as:(7)$$j=[j_{1}^{T},j_{2}^{T},\cdots j_{M}^{T}],$$
where $j_{m}=\sum_{n=1}^{N_{j}}a_{R}\left(\theta_{j,n}\right)⊗s_{j,n,m}$ denotes the return signal of jamming at the *m*th pulse, $s_{j,n,m}\in {}^{L\times 1}$ contains the *n*th jamming signal, $N_{j}$ is the total number of jamming, $\theta_{j,n}$ is the direction of the *n*th jamming.

So the jamming covariance matrix is written as:(8)$$\begin{array}{c} {\tilde{R}}_{j}=E\left[{\mathrm{jj}}_{}^{H}\right]=I_{M}⊗R_{j}⊗I_{L}\\ R_{j}=\sum_{n=1}^{N_{c}}\sigma_{j,n}^{2}a_{R}\left(\theta_{j,n}\right)a_{R}^{T}\left(\theta_{j,n}\right). \end{array}$$

### 2.4. Noise

Noise is usually considered as additive white Gaussian noise, its power is expressed as:(9)$$\sigma^{2}=kbT_{0}BF,$$
where $kb=1.38\times 10^{-23}$ J/K is the Boltzmann constant, $T_{0}$ is the noise temperature, which is usually taken as 290 K, *B* is the equivalent sampling bandwidth, *F* is the noise coefficient. During the simulation process, the noise signal is generated by a zero-mean complex Gaussian random distribution which takes the noise power as the variance.

The noise covariance matrix is written as:(10)$${\tilde{R}}_{n}=E\left[nn^{H}\right]=\sigma^{2}I_{MN_{R}L}.$$

## 3. Review of Transceiver Joint Optimization with Fixed Pulse-to-Pulse Waveform

The joint optimization with fixed pulse-to-pulse waveform is the basis of joint optimization with pulse-to-pulse variations. This is a joint design of transmit waveforms and receive filters for MIMO radar systems. The aim is to maximize the output SINR so that to achieve enhanced detection performance for slow-moving targets that might be obscured by clutter and jamming [2]. Here we review the basic joint optimization under energy constraint.

Assume that $s_{n}\left(t\right)$ denotes the transmit waveform of the *n*th transmitter, within a coherent processing interval (CPI) which contains *M* pulse, the transmit waveform of the *n*th transmitter is expressed as:(11)$${\tilde{s}}_{n}\left(t\right)=\sum_{m=1}^{M}s_{n}\left(t-MT_{r}\right),$$
in which $T_{r}=1/f_{r}$ denotes the pulse repetition interval (PRI) and $f_{r}$ denotes the pulse repetition frequency (PRF). Assume that the waveform matrix of the system is expressed as $S={\left[s_{1},\cdots ,s_{N_{T}}\right]}^{T}\in C^{N_{T}\times L}$, where $s_{n}$ denotes the digital sampling of $s_{n}\left(t\right)$ .

The transceiver joint optimization under maximum output SINR criterion is aimed at optimize both the transmit waveform and the receive filters at the same time, so as to get the maximal output SINR.

Let $w={\left[w_{1}^{T},w_{2}^{T},\cdots ,w_{N_{R}}^{T}\right]}^{T}$ denotes the receive filters, in which $w_{j}\in C^{ML\times 1},j=1,2,\cdots ,N_{R}$ denotes the *j*th receive filter. The objective function of the maximum output SINR is written as:(12)$$\begin{array}{c} \max_{w,S}SINR\left(w,S\right)=\frac{{\left|\alpha_{t}\right|}^{2}{\left|w^{H}{\tilde{S}}^{T}v_{t}\right|}^{2}}{w^{H}\left({\tilde{R}}_{c}+{\tilde{R}}_{jn}\right)w}\;,\\ s.t.tr\left(SS^{H}\right)=Me_{t}. \end{array}$$
where $\tilde{S}=I_{M}⊗I_{N_{R}}⊗S$, $v_{t}=u\left(f_{d}\right)⊗a_{R}\left(\theta_{t}\right)⊗a_{T}\left(\theta_{t}\right)$, $u\left(f_{d}\right)={\left[1,\cdots ,e^{j2\pi \left(M-1\right)f_{d}T_{r}}\right]}^{T}$ denotes the temporal steering vector. ${\tilde{R}}_{jn}={\tilde{R}}_{j}+{\tilde{R}}_{n}$, $e_{t}$ denotes the total energy, $tr\left(SS^{H}\right)=Me_{t}$ is the energy constraint.

During the cyclic optimization process, we always optimize the waveform with fixed receive filters and optimize the receive filters with fixed waveform within one iteration, continue this cycle until the output SINR contract to maximum, so as to get good clutter suppressing results.

The derivation process is similar to Reference [2], to jointly optimize the transceiver with fixed pulse-to-pulse waveform, while optimizing the receive filters with fixed waveform, regard (11) as a generalized Rayleigh quotient, then we get the optimal solution of receive filters:(13)$$w_{opt}={\left({\tilde{R}}_{c}+{\tilde{R}}_{jn}\right)}^{-1}{\tilde{S}}^{T}v_{t}.$$

When optimize waveform with fixed receiver filters, the optimal solution of waveform is:(14)$$s_{opt}=\sqrt{Me_{t}}h*\left(W\right)/{∥h(W)∥}_{2}$$
where $h\left(W\right)={\tilde{R}}_{DL}^{-1}\left(W\right)\left(W*⊗I_{N_{T}}\right)v_{t}$, ${\tilde{R}}_{DL}\left(W\right)={\tilde{R}}_{c}\left(W\right)+\beta \left(W\right)I_{LN_{T}}$, $\beta \left(W\right)=w^{H}{\tilde{R}}_{jn}w/e_{t}$, refer to Reference [2] for detailed deduction.

The whole optimization process can be summarized as the following steps: 

**Step 1**: *n* = 0, initialize the waveform S(n);

**Step 2**: *n* = *n* + 1, compute the clutter covariance matrix ${\tilde{R}}_{c}$ using Equation (Equation 6), compute the optimal receive filter $w^{\left(n\right)}$ using Equation (Equation 13);

**Step 3**: Compute ${\tilde{R}}_{c}\left(W^{\left(n\right)}\right)$, ${\tilde{R}}_{DL}\left(W^{\left(n\right)}\right)$ and $t(W^{\left(n\right)})$, then compute the optimal waveform of this iteration using Equation (Equation 14);

**Step 4**: Repeat step 2 and step 3, until the output SINR converges. 

As to the computational complexity of transceiver joint optimization with fixed pulse-to-pulse waveform under energy constraint, it is linear w.r.t. the number of iterations and the complexity involved in each iteration. At each iteration, the optimization of w (with (13)) requires $O({\left(LMN_{R}\right)}^{3})$ operations and the optimization of S (with (14)) requires $O({\left(LN_{T}\right)}^{3})$ operations.

## 4. Transceiver Joint Optimization with Pulse-to-Pulse Waveform Variations

The joint optimization with pulse-to-pulse waveform variations is generalized form joint optimization with fixed pulse-to-pulse waveform.

To jointly optimize the transceiver with pulse-to-pulse waveform variations, let $S={\left[s_{1,m},s_{2,m},\cdots s_{N_{T},m}\right]}^{T}\in C^{N_{T}\times L}$ denotes the system’s waveform matrix at the *m*th pulse, in which $s_{n,m}$ denotes the digital sampling of the transmit waveform form the *n*th transmit antennas at the *m*th pulse. Then the return signal of the target direction at the *m*th pulse is written as:(15)$$T_{m}=\alpha_{t}e^{j2\pi \left(m-1\right)f_{d}T_{r}}a_{R}\left(\theta_{t}\right)a_{T}^{T}\left(\theta_{t}\right)S_{m}.$$

The vectored result of $T_{m}$ is:(16)$$t_{m}=\mathrm{vec}\left(T_{m}\right)=\alpha_{t}e^{j2\pi \left(m-1\right)f_{d}T_{r}}\left(I_{L}⊗A\left(\theta_{t}\right)\right){\hat{s}}_{m},$$
where $A\left(\theta_{t}\right)=a_{R}\left(\theta_{t}\right)a_{T}^{T}\left(\theta_{t}\right)$, ${\hat{s}}_{m}=\mathrm{vec}\left(S_{m}\right)$.

Assume that $T=[t_{1},t_{2},\cdots t_{M}]$, $\hat{S}=\left[{\hat{s}}_{1},{\hat{s}}_{2},\cdots {\hat{s}}_{M}\right]$, $D\left(f_{d}\right)=\mathrm{diag}\left(\left[1,\cdots ,e^{j2\pi \left(M-1\right)f_{d}T_{r}}\right]\right)$, $\hat{s}=\mathrm{vec}\left(\hat{S}\right)$, t=vec(T), then we get:(17)$$T=\alpha_{t}\left(I_{L}⊗A\left(\theta_{t}\right)\right)\hat{S}D\left(f_{d}\right).$$

And the vectored result is:(18)$$t=\alpha_{t}\left(D\left(f_{d}\right)⊗I_{L}⊗A\left(\theta_{t}\right)\right)\hat{s}.$$

Use a compute process similar to Section 3, the vectored result of clutter return signal is:(19)$$c=\sum_{l=-K}^{K}\sum_{k=1}^{N_{c}}\alpha_{c,l,k}\left(D\left(f_{c,l,k}\right)⊗P_{-l}⊗A\left(\theta_{c,l,k}\right)\right)\hat{s}.$$

In this way, the clutter covariance matrix is written as:(20)$${\tilde{R}}_{c}\left(\hat{s}\right)=\sum_{l=-K}^{K}\sum_{k=1}^{N_{c}}\sigma_{c,l,k}^{2}\left(D\left(f_{c,l,k}\right)⊗P_{-l}⊗A\left(\theta_{c,l,k}\right)\right)\hat{s}{\hat{s}}^{H}\left(D^{H}\left(f_{c,l,k}\right)⊗P_{l}⊗A^{H}\left(\theta_{c,l,k}\right)\right).$$

Compared with the transceiver joint optimization with fixed pulse-to-pulse waveform, when the optimization is generalized to condition with pulse-to-pulse waveform variations, as it increases a changing dimension, the contract speed will be slower and the computational complexity will increase. But the output SINR reaches a relatively higher level, so the clutter suppressing performance will also be superior and the processing result will be better accordingly. Its advantage also reflected in achieving a lower minimum detectable velocity (MDV), that is important in many practical applications. Also, this method will be meaningful of the multi-target detection and the design of a more robust detection algorithm.

Based on the computations above, with pulse-to-pulse waveform variations, the output SINR is written as:(21)$$SINR\left(w,\hat{s}\right)=\frac{{\left|\alpha_{t}\right|}^{2}{\left|w^{H}\left(D\left(f_{d}\right)⊗I_{L}⊗A\left(\theta_{t}\right)\right)\hat{s}\right|}^{2}}{w^{H}\left({\tilde{R}}_{c}\left(\hat{s}\right)+{\tilde{R}}_{jn}\right)w}.$$

### 4.1. Joint Optimization under Energy Constraint

Firstly, we discuss the optimization under energy constraint which is a basic constraint of the transceiver joint optimization. The objective function of the transceiver joint optimization with pulse-to-pulse waveform variations is written as:(22)$$\begin{array}{c} \max_{w,\hat{s}}\frac{{\left|w^{H}\left(D\left(f_{d}\right)⊗I_{L}⊗A\left(\theta_{t}\right)\right)\hat{s}\right|}^{2}}{w^{H}\left({\tilde{R}}_{c}\left(\hat{s}\right)+{\tilde{R}}_{jn}\right)w},\\ s.t.{\hat{s}}^{H}\hat{s}=Me_{t}. \end{array}$$

Similar to the optimization with fixed pulse-to-pulse waveform, cyclic optimization process is also used in this situation. During the optimization, while optimizing the receive filters with fixed waveform, regard (21) as a generalized Rayleigh quotient, then we get the optimal receive filters:(23)$$w_{opt}={\left({\tilde{R}}_{c}\left(\hat{s}\right)+{\tilde{R}}_{jn}\right)}^{-1}\left(D\left(f_{d}\right)⊗I_{L}⊗A\left(\theta_{t}\right)\right)\hat{s}.$$

When optimizing the waveform with fixed receiver filters, the optimal solution of waveform is:(24)$${\hat{s}}_{opt}=\sqrt{Me_{t}}g\left(W\right)/{∥g(W)∥}_{2}$$
where $g\left(W\right)={\left({\tilde{R}}_{c}\left(w\right)+\gamma \left(w\right)I_{MLN_{T}}\right)}^{-1}\left(D^{H}\left(f_{d}\right)⊗I_{L}⊗A^{H}\left(\theta_{t}\right)\right)w$, $\gamma \left(w\right)=w^{H}{\tilde{R}}_{jn}w/Me_{t}$, ${\tilde{R}}_{c}\left(w\right)=\sum_{l=-K}^{K}\sum_{k=1}^{N_{c}}\sigma_{c,l,k}^{2}\left(D^{H}\left(f_{c,l,k}\right)⊗P_{l}⊗A^{H}\left(\theta_{c,l,k}\right)\right)ww^{H}\left(D\left(f_{c,l,k}\right)⊗P_{-l}⊗A\left(\theta_{c,l,k}\right)\right)$ .

Cyclicly optimize the transmit waveform and receive filters using (22) and (23), let $SINR^{\left(n\right)}$ denotes the output SINR after the *n*th iteration. According to the practical need, set a threshold value ε,ε>0, if we have:(25)$$\frac{\left|SINR^{\left(n\right)}-SINR^{\left(n-1\right)}\right|}{SINR^{\left(n-1\right)}}<\varepsilon .$$

Then the output SINR has contracted to maximum, the results we get are considered as the optimal transmit waveform and receive filters. As the output SINR is at its maximum at this time, the clutter suppressing performance will be superior.

The whole optimization process can be summarized as the following steps: 

**Step 1**: *n* = 0, initialize the waveform S(n);

**Step 2**: *n* = *n* + 1, compute the clutter covariance matrix ${\tilde{R}}_{c}$ using Equation (Equation 20), compute the optimal receive filter $w^{\left(n\right)}$ using Equation (Equation 23);

**Step 3**: Compute ${\tilde{R}}_{c}\left(W^{\left(n\right)}\right)$, ${\tilde{R}}_{DL}\left(W^{\left(n\right)}\right)$ and $t(W^{\left(n\right)})$, then compute the optimal waveform of this iteration using Equation (Equation 24);

**Step 4**: Repeat step 2 and step 3, until the output SINR converges.

### 4.2. The Addition of Constant-Envelope Constraint

During the MIMO GMTI Radar waveform optimization, constraints of the transmit waveform are always been considered, so as to make the transmit waveform meet some practical requirements. For example, consider the cost of the transmitter and the affordability of the transmit antennas, we need to constrain the total transmit energy; in order to avoid the distortion of the transmit waveform due to power amplification, constant-envelope waveforms are usually used; and the similarity constraint can control the shape of the ambiguity function of the waveform and avoid the drawbacks of the waveforms under energy constraint and constant-envelope constraint. In addition, there are constraints such as peak-to-average ratio [22] and spectrum compatibility.

In the discussions in Section 4.2 and Section 4.3, we not just focus on the clutter suppression performance of the algorithms but also consider when add these constraints, apply our joint optimization methods, how the overall performance of the system will change.

Constant-envelope waveforms are usually used in practice, in order to make the amplifier work at maximum efficiency and avoid the unnecessary non-linear effects of the transmitter.

In this subsection, based on the optimization under energy constraint, we add constant-envelope constraint to the optimization and the objective function is written as:(26)$$\begin{array}{c} \max_{w,\hat{s}}\frac{{\left|w^{H}\left(D\left(f_{d}\right)⊗I_{L}⊗A\left(\theta_{t}\right)\right)\hat{s}\right|}^{2}}{w^{H}\left({\tilde{R}}_{c}\left(\hat{s}\right)+{\tilde{R}}_{jn}\right)w},\\ s.t.\;\;\left|s\left(i\right)\right|=\sqrt{p_{s}},i=1,\cdots ,N_{T}L, \end{array}$$
in which $p_{s}=Me_{t}/N_{T}L$ .

#### 4.2.1. Optimization Based on Relaxation and Randomization

As the optimization problem under constant-envelope constraint is non-convex, we consider the optimization based on relaxation and randomization [31]. We also solve it with convergence guarantee resorting to Reference [35]. In particular, we may optimize one phase at a time as well as the receive filter in either a cyclic way or resorting to the MBI.

The objective function (25) is written as:(27)$$\begin{array}{c} \mathrm{SINR}\left(w,R_{s}\right)=\frac{{\left|w^{H}\left(D\left(f_{d}\right)⊗I_{L}⊗A\left(\theta_{t}\right)\right)\hat{s}\right|}^{2}}{w^{H}\left({\tilde{R}}_{c}\left(\hat{s}\right)+{\tilde{R}}_{jn}\right)w}\\ =\frac{w^{H}K\left(R_{s}\right)w}{w^{H}\left({\tilde{R}}_{c}\left(\hat{s}\right)+{\tilde{R}}_{jn}\right)w}\\ =\frac{\mathrm{tr}[X\left(w\right)R_{s}]}{\mathrm{tr}[R_{DL}\left(w\right)R_{s}]}, \end{array}$$
in which $R_{s}=s^{*}s^{T}$, $X\left(w\right)=\left(D\left(f_{d}\right)⊗I_{L}⊗A\left(\theta_{t}\right)\right)ww^{H}{\left(D\left(f_{d}\right)⊗I_{L}⊗A\left(\theta_{t}\right)\right)}^{H}$, $K\left(R_{s}\right)=\left(D\left(f_{d}\right)⊗I_{L}⊗A\left(\theta_{t}\right)\right)R_{s}{\left(D\left(f_{d}\right)⊗I_{L}⊗A\left(\theta_{t}\right)\right)}^{H}$, ${\tilde{R}}_{DL}\left(w\right)={\tilde{R}}_{c}\left(w\right)+\gamma \left(w\right)I_{MLN_{T}}$.

Using formulation (26), we reformulate the optimization problem (25) as following:(28)$$\begin{array}{c} \underset{w,R_{s}}{max}\;\mathrm{SINR}\left(w,R_{s}\right)\\ s.t.\;\;\mathrm{diag}\left(R_{s}\right)=p_{s}\cdot 1,\;\mathrm{rank}\left(R_{s}\right)=1,\;R_{s}⪰0. \end{array}$$

Here, we use relaxation process firstly to tackle the rank constraint of (27) by drop the rank constraint of $R_{s}$, then we get the associated relaxed problem:(29)$$\begin{array}{c} \underset{w,R_{s}}{max}\;\mathrm{SINR}\left(w,R_{s}\right)\\ s.t.\;\;\mathrm{diag}\left(R_{s}\right)=p_{s}\cdot 1,\;\;R_{s}⪰0. \end{array}$$

Similar to the optimization under energy constraint, cyclic optimization process is also used in this situation. During the optimization, while optimizing the receive filters with fixed transmit waveform, consider the objective function as:(30)$$\mathrm{SINR}\left(w,R_{s}\right)=\frac{w^{H}K\left(R_{s}\right)w}{w^{H}\left({\tilde{R}}_{c}\left(\hat{s}\right)+{\tilde{R}}_{jn}\right)w}.$$

Regard (29) as a generalized Rayleigh quotient, then we get the optimal solution of receive filters:(31)$$w^{opt}={\tilde{R}}_{u}^{-1/2}P\left({\tilde{R}}_{u}^{-1/2}K\left(R_{s}\right){\tilde{R}}_{u}^{-1/2}\right),$$
in which, ${\tilde{R}}_{u}={\tilde{R}}_{c}\left(R_{s}\right)+{\tilde{R}}_{jn}$ and $P\left({\tilde{R}}_{u}^{-1/2}K\left(R_{s}\right){\tilde{R}}_{u}^{-1/2}\right)$ denotes the principal eigenvector associated with the largest eigenvalue of matrix ${\tilde{R}}_{u}^{-1/2}K\left(R_{s}\right){\tilde{R}}_{u}^{-1/2}$.

While optimizing the transmit waveform with fixed receive filters, consider the objective function as:(32)$$\begin{array}{c} \underset{R_{s}}{max}\frac{\mathrm{tr}\left[X\left(w\right)R_{s}\right]}{\mathrm{tr}\left[{\tilde{R}}_{DL}\left(W\right)R_{s}\right]}\\ s.t.\;\;\mathrm{diag}\left(R_{s}\right)=p_{s}\cdot 1,\;\;R_{s}⪰0. \end{array}$$

As the problem (31) is quasiconcave, according to the Charnes-Cooper transform, we consider the following semi-definite programming (SDP) to resolve the linear fractional programming problem above:(33)$$\begin{array}{c} \max_{M,t}\;\mathrm{tr}\left[X\left(W,v_{t}\right)M\right]\\ s.t.\;\;\mathrm{tr}\left[{\tilde{R}}_{DL}\left(W\right)M\right]=1,\;\mathrm{diag}\left(M\right)=tp_{s}\cdot 1,\;\;M⪰0. \end{array}$$

The optimization problem above is convex and the optimized solution is written as:(34)$$R_{s}^{opt}=M^{opt}/t^{opt}.$$

Cyclicly optimize the transmit waveform and receive filters until the output SINR grows less than the predetermined threshold . Let $R_{s}^{*}$ be the final waveform matrix after the circulation is over. If this $R_{s}^{*}$ is of rank 1, that is, $R_{s}^{*}=s^{*}{\left(s^{*}\right)}^{H}$, then we get the constant-envelope waveform is gotten immediately.

If this $R_{s}^{*}$ is not of rank 1, use the randomization process proposed in Reference [36] to get the constant-envelope waveform form $R_{s}^{*}$ . During the randomization process, generate *r* random vectors: $e_{1},e_{2},\cdots ,e_{r}$, which are with a circular symmetric complex Gaussian distribution with the mean of 0 and the variance of 1. Define $v_{k}=\sqrt{p_{s}}\exp\left(j∠e_{k}\right),k=1,2,\cdots ,r$, compute:(35)$$\mathrm{SINR}\left(y_{k}\right)=\frac{{\left|y_{k}^{T}\left(D\left(f_{d}\right)⊗I_{L}⊗A\left(\theta_{t}\right)\right)w\right|}^{2}}{y_{k}^{T}\left({\tilde{R}}_{c}\left(w\right)+\gamma \left(w\right)I_{MLN_{T}}\right)y_{k}^{*}},k=1,\cdots r.$$

Then the constant-envelope waveform is:(36)$$s=\arg\max_{y_{k}}\;\mathrm{SINR}\left(y_{k}\right),$$
where $\arg\max_{y_{k}}$ is the set of all $y_{k}$ that maximize the expression above.

The whole optimization process can be summarized as the following steps: 

**Step 1**: *n* = 0, initialize the waveform S(n);

**Step 2**: *n* = *n* + 1, compute K(Rs), compute the optimal receive filter w(n) using Equation (Equation 31);

**Step 3**: Compute R˜c(W(n)), R˜DL(W(n)) and X(w(n));

**Step 4**: Solve the semi-definite programming of Equation (Equation 33), find the optimal $\left\{M^{\left(n\right)},t^{\left(n\right)}\right\}$, then ${R_{s}}^{\left(n\right)}=M^{\left(n\right)}/t^{\left(n\right)}$;

**Step 5**: Repeat step 2 and step 3, until the output SINR converges, then get the constant-envelope waveform using Equation (Equation 36).

#### 4.2.2. Optimization Based on Fractional Programming and Power-Like Iteration

In this section, an optimization based on the method of fractional programming and power-like iteration is used to deal with the non-convex optimization problem.

In this method, optimize the receive filters with the fixed transmit waveforms using the previous design method, also use the maximization output SINR as the criterion. The following is focused on the process of designing constant-envelope waveform with fixed receive filters.

The objective function of the constant-envelope waveform design can be re-represented as:(37)$$\begin{array}{c} \max_{s}\frac{s^{T}X\left(w\right)s^{*}}{s^{T}{\tilde{R}}_{DL}\left(w\right)s^{*}}\\ s.t.\;\;\left|s\left(i\right)\right|=\sqrt{p_{s}},i=1,\cdots ,N_{T}L. \end{array}$$

The problem is then addressed using the fractional programming approach and the proposed method involves iterative process.

Let $s^{\left(n,k\right)}$ be the waveform in the (n,k)th iteration, where *n* denotes the ordinal of the outer iteration and *k* denotes the ordinal of the inner iteration, $f^{\left(n,k\right)}$ representing the corresponding target value of $s^{\left(n,k\right)}$, then the optimization problem in the (n,k+1)th iteration can be expressed as:(38)$$\begin{array}{c} \max_{s}s^{T}\left[X\left(w^{\left(n\right)}\right)-f^{\left(n,k\right)}{\tilde{R}}_{DL}\left(w^{\left(n\right)}\right)\right]s^{*}\\ s.t.\;\;\left|s\left(i\right)\right|=\sqrt{p_{s}},i=1,\cdots ,N_{T}L. \end{array}$$

Let $Q^{\left(n,k\right)}=X\left(w^{\left(n\right)}\right)-f^{\left(n,k\right)}{\tilde{R}}_{DL}\left(w^{\left(n\right)}\right)$, $T^{\left(n,k\right)}=Q^{\left(n,k\right)}+\mu I$, where μ is a guaranteed constant. It is easy to prove that the formula (37) is equivalent to the following objective function:(39)$$\max_{s}s^{T}T^{\left(n,k\right)}s^{*},\;\;s.t.\;\;\left|s\left(i\right)\right|=\sqrt{p_{s}},i=1,\cdots ,N_{T}L.$$

Set the initial point $s^{\left(n,k\right)}$ and the algorithm converges at a local maximum or saddle point. Let $g\left(s\right)$ represent the objective function of expression (37), $s^{\left(n,k+1\right)}$ represent the solution of the iterative method, so there are:(40)$$g\left(s^{\left(n,k+1\right)}\right)\geq g\left(s^{\left(n,k\right)}\right)=0.$$

And there are:(41)$$f^{\left(n,k+1\right)}=\frac{{\left(s^{\left(n,k+1\right)}\right)}^{T}X\left(w^{\left(n\right)}\right){\left(s^{\left(n,k+1\right)}\right)}^{*}}{{\left(s^{\left(n,k+1\right)}\right)}^{T}{\tilde{R}}_{DL}\left(w^{\left(n\right)}\right){\left(s^{\left(n,k+1\right)}\right)}^{*}}\geq f^{\left(n,k\right)}$$

Therefore, the method is convergent, since $f^{\left(n,k\right)}$ is not decrementally associated with *k*.

The whole optimization process can be summarized as the following steps: 

**Step 1**: *n* = 0, initialize the waveform S(n);

**Step 2**: *n* = *n* + 1, compute K(Rs), compute the optimal receive filter w(n) using Equation (Equation 31);

**Step 3**: Compute R˜c(W(n)), R˜DL(W(n)) and X(w(n)). k=0, let sn,0=vec(S(n−1)) and fn,0 denotes the objective function value corresponding to sn,0;

**Step 4**: Compute $Q^{\left(n,k\right)}$ and $T^{\left(n,k\right)}$, then get the optimal waveform that satisfy Equation (Equation 39);

**Step 5**: *k* = *k* + 1, repeat step 4 until f(n,k)−f(n,k−1)/f(n,k)<δ,δ>0;

**Step 6**: Repeat steps 2 to 4, until the output SINR converges.

### 4.3. The Addition of Similarity Constraint

In this section, we consider to add similarity constraint to the joint optimization of the transmit waveform and receive filters. The similarity constraint controls the shape of the ambiguity function of the waveform and avoid the drawbacks of the waveforms under energy constraint and constant-envelope constraint.

Let $s_{0}$ to be the reference waveform, which has good features like high range resolution, low side-lobe and constant envelope. The similarity constraint is written as:(42)$${∥s-s_{0}∥}_{2}^{2}\leq \delta ,$$
in which δ is a threshold value that users selected to rule the size of similarity region and $0\leq \delta \leq 2e_{t}$.

Generalize the algorithm above to similarity constrained waveform and the optimization problem is written as:(43)$$\begin{array}{c} \max\left\{\frac{{\left|{\hat{s}}^{T}\left(D\left(f_{d}\right)⊗I_{L}⊗A\left(\theta_{t}\right)\right)w\right|}^{2}}{{\hat{s}}^{T}\left({\tilde{R}}_{c}\left(w\right)+\gamma \left(w\right)I_{MLN_{T}}\right){\hat{s}}^{*}}\right\}\\ s.t.\;\;s^{T}s^{*}=Me_{t},\;\;\;{∥s-s_{0}∥}_{2}^{2}\leq \delta . \end{array}$$

Here, cyclic optimization process is also used in this situation. Regard (37) as a generalized Rayleigh quotient, then we get the optimal solution of receive filters [32]:(44)$$w_{opt}={\left({\tilde{R}}_{c}\left(\hat{s}\right)+{\tilde{R}}_{jn}\right)}^{-1}\left(D\left(f_{d}\right)⊗I_{L}⊗A\left(\theta_{t}\right)\right)\hat{s}.$$

#### 4.3.1. Optimization Based on Relaxation and Rank-One Decomposition

To optimize the transmit waveform, we also choose relaxation and rank-one decomposition to deal with this problem.

The similarity constraint in (37) is reformulate as $s_{0}^{H}ss^{H}s_{0}\geq \eta$ and $\eta ={\left(e_{t}-\delta /2\right)}^{2}$. So the objective function (37) is reformulate as:(45)$$\begin{array}{c} \max\left\{\frac{{\left|{\hat{s}}^{T}\left(D\left(f_{d}\right)⊗I_{L}⊗A\left(\theta_{t}\right)\right)w\right|}^{2}}{{\hat{s}}^{T}\left({\tilde{R}}_{c}\left(w\right)+\gamma \left(w\right)I_{MLN_{T}}\right){\hat{s}}^{*}}\right\}\\ s.t.\;\;s^{T}s^{*}=e_{t},\;\;\;s_{0}^{H}ss^{H}s_{0}\geq \eta . \end{array}$$

Objective function (39) is equivalent to the following objective function:(46)$$\begin{array}{c} \underset{R_{s}}{max}\frac{\mathrm{tr}\left[X\left(w\right)R_{s}\right]}{\mathrm{tr}\left[{\tilde{R}}_{DL}\left(W\right)R_{s}\right]}\\ s.t.\;\;\mathrm{tr}\left(R_{s}\right)=e_{t},\mathrm{tr}\left(R_{s}R_{0}\right)\geq \eta ,R_{s}\geq 0,\mathrm{rank}\left(R_{s}\right)=1, \end{array}$$
where $R_{0}=s_{0}^{T}s_{0}^{*}$. Here, we also use relaxation process to tackle the rank constraint of (40) by drop the rank constraint of $R_{s}$, then we get the associated relaxed problem:(47)$$\begin{array}{c} \underset{R_{s}}{max}\frac{\mathrm{tr}\left[X\left(w\right)R_{s}\right]}{\mathrm{tr}\left[{\tilde{R}}_{DL}\left(W\right)R_{s}\right]}\\ s.t.\;\;\mathrm{tr}\left(R_{s}\right)=e_{t},\mathrm{tr}\left(R_{s}R_{0}\right)\geq \eta ,R_{s}\geq 0. \end{array}$$

Similar to the solution process in Section 4.2, use the Charnes-Cooper transform, we solve the following SDP to get the solution of (41):(48)$$\begin{array}{c} \max_{M_{sim},\varsigma }\;\mathrm{tr}\left[X\left(w\right)M_{sim}\right]\\ s.t.\;\;\mathrm{tr}\left[{\tilde{R}}_{DL}\left(w\right)M_{sim}\right]=1,\;\mathrm{tr}\left(M_{sim}\right)=\varsigma e_{t},\;\;\mathrm{tr}\left(M_{sim}R_{0}\right)=\eta \varsigma ,M_{sim}\geq 0. \end{array}$$

The optimal solution of (42) is expressed as $\left\{M_{sim}^{opt},\varsigma^{opt}\right\}$, then we get:(49)$$R_{s}^{opt}=M_{sim}^{opt}/\varsigma^{opt}.$$

If $R_{s}^{opt}$ is rank 1, we get the optimal transmit waveform directly; otherwise, use the matrix decomposition theorem in Reference [36] to extract the similarity-constrained waveform.

If $\mathrm{rank}\left(R_{s}^{opt}\right)\geq 2$, we find a vector $s^{opt}$ which meet the following equation:(50)$$\begin{array}{c} \left(\mathrm{tr}\left(R_{s}^{opt}X\left(w\right)\right),\mathrm{tr}\left(R_{s}^{opt}R_{DL}\left(w\right)\right),\mathrm{tr}\left(R_{s}^{opt}I_{MN_{T}L}\right),\mathrm{tr}\left(R_{s}^{opt}R_{0}\right)\right)\\ =\left({\left(s^{opt}\right)}^{H}X\left(w\right)s^{opt},{\left(s^{opt}\right)}^{H}R_{DL}\left(w\right)s^{opt},{\left(s^{opt}\right)}^{H}s^{opt},{\left(s^{opt}\right)}^{H}R_{0}s^{opt}\right). \end{array}$$

Therefore, we get:(51)$$\begin{array}{c} \mathrm{tr}\left(s^{opt}{\left(s^{opt}\right)}^{H}\right)=\mathrm{tr}\left(R_{s}^{opt}\right)=e_{t},\\ \mathrm{tr}\left(s^{opt}{\left(s^{opt}\right)}^{H}R_{0}\right)=\mathrm{tr}\left(R_{s}^{opt}R_{0}\right)>\eta ,s^{opt}{\left(s^{opt}\right)}^{H}\geq 0. \end{array}$$

So that $s^{opt}{\left(s^{opt}\right)}^{H}$ is feasible for (41) and:(52)$$\frac{\mathrm{tr}\left[X\left(w\right)s^{opt}{\left(s^{opt}\right)}^{H}\right]}{\mathrm{tr}\left[{\tilde{R}}_{DL}\left(w\right)s^{opt}{\left(s^{opt}\right)}^{H}\right]}=\frac{\mathrm{tr}\left[X\left(w\right)R_{s}^{*}\right]}{\mathrm{tr}\left[{\tilde{R}}_{DL}\left(w\right)R_{s}^{*}\right]}.$$

So $s^{opt}{\left(s^{opt}\right)}^{H}$ is the optimal rank 1 solution and $s^{opt}$ is the optimal solution of (39).

The whole optimization process can be summarized as the following steps: 

**Step 1**: *n* = 0, initialize the waveform S(n);

**Step 2**: *n* = *n* + 1, compute ${\tilde{R}}_{c}$, compute the optimal receive filter $w^{\left(n\right)}$ using Equation (Equation 31);

**Step 3**: Compute R˜c(W(n)), R˜DL(W(n)) and X(w(n));

**Step 4**: Solve the semi-definite programming of Equation (Equation 33), find the optimal $\left\{{M_{sim}}^{opt},\zeta^{opt}\right\}$, then ${R_{s}}^{opt,(n)}={M_{sim}}^{opt,(n)}/\zeta^{opt,(n)}$;

if $\mathrm{rank}({R_{s}}^{opt,(n)})=1$, get the optimal waveform ${\overset{⌣}{\;s}}^{\left(n\right)}$directly; if $\mathrm{rank}({R_{s}}^{opt,(n)})=2$, ${\overset{⌣}{\;s}}^{\left(n\right)}$ = D2(**X**(**w**(n)),**R**DL(**w**(n)),**I**,**R**0);

if $\mathrm{rank}({R_{s}}^{opt,(n)})\geq 3$, ${\overset{⌣}{\;s}}^{\left(n\right)}$ = D1(**X**(**w**(n)),**R**DL(**w**(n)),**I**,**R**0);

**Step 5**: Repeat step 2 to step 4, until the output SINR converges.

#### 4.3.2. Optimization Based on Fractional Programming and the SWORD Method

For the non-convex optimization problem represented by the formula (42), the design idea based on the fractional programming and the SWORD method is also utilized.

Its iterative approach is similar to Section 4.2.2, which solves the following problems in the (n,k)th iteration:(53)$$\begin{array}{c} \max_{s}s^{T}\left[X\left(w^{\left(n\right)}\right)-f^{\left(n,k\right)}{\tilde{R}}_{DL}\left(w^{\left(n\right)}\right)\right]s^{*}\\ s.t.\;\;s^{T}s^{*}=e_{t},\;\;\;{∥s-s_{0}∥}_{2}^{2}\leq \delta . \end{array}$$

Using the same concept in Section 4.2.2, the optimization problem of processing (52) is equivalent to processing:(54)$$\max_{s}s^{T}T^{\left(n,k\right)}s^{*},\;\;s.t.\;\;s^{T}s^{*}=e_{t},\;\;\;{∥s-s_{0}∥}_{2}^{2}\leq \delta .$$

Its optimal solution is:(55)$$s^{\left(n,k+1\right)}=\lambda_{1}^{\left(n,k+1\right)}{\left(\lambda_{2}^{\left(n,k+1\right)}I-{\left(T^{\left(n,k\right)}\right)}^{*}\right)}^{-1}s_{0}.$$

Among them $\lambda_{1}^{\left(n,k+1\right)}=\left(e_{t}-\delta /2\right)/\left(s_{0}^{H}{\left(\lambda_{2}^{\left(n,k+1\right)}I-{\left(T^{\left(n,k\right)}\right)}^{*}\right)}^{-1}s_{0}\right)$, $\lambda_{2}^{\left(n,k+1\right)}$ is the solution of the following formula:(56)$$\frac{s_{0}^{H}{\left(\lambda_{2}^{}I-{\left(T^{\left(n,k\right)}\right)}^{*}\right)}^{-2}s_{0}}{{\left[s_{0}^{H}{\left(\lambda_{2}^{}I-{\left(T^{\left(n,k\right)}\right)}^{*}\right)}^{-1}s_{0}\right]}^{2}}=\frac{e_{t}}{{\left(e_{t}-\delta /2\right)}^{2}}.$$

The whole optimization process can be summarized as the following steps: 

**Step 1**: *n* = 0, initialize the waveform S(n);

**Step 2**: *n* = *n* + 1, compute K(Rs), compute the optimal receive filter w(n) using Equation (Equation 31);

**Step 3**: Compute R˜c(W(n)), R˜DL(W(n)) and X(w(n)). k=0, let sn,0=vec(S(n−1)) and fn,0 denotes the objective function value corresponding to sn,0;

**Step 4**: Compute $Q^{\left(n,k\right)}$ and $T^{\left(n,k\right)}$, then get the optimal waveform using Equation (Equation 55);

**Step 5**: *k* = *k* + 1, repeat step 4 until f(n,k)−f(n,k−1)/f(n,k)<δ,δ>0;

**Step 6**: Repeat steps 2 to 4, until the output SINR converges.

### 4.4. Discussion of Computational Complexity

As to the computational complexity of transceiver joint optimization with fixed pulse-to-pulse waveform under energy constraint, the discussion is similar to the optimization with fixed pulse-to-pulse waveforms. For each iteration, the optimization of w (with (13)) requires $O({\left(LMN_{R}\right)}^{3})$ operations and the optimization of S (with (14)) requires $O({\left(LMN_{T}\right)}^{3})$ operations.

For the computational complexity of both transceiver joint optimization algorithms under constant-envelope constraint, we focus on the complexity involved in each (outer) iteration. For optimization based on relaxation and randomization, the optimization of w requires $O({\left(LMN_{R}\right)}^{3})$ operations and the optimization of $R_{s}$ through solving the SDP requires $O({\left(N_{T}ML\right)}^{4.5})$ operations. Optimization based on fractional programming and power-like iteration involves $O({\left(LMN_{R}\right)}^{3})$ operations to optimize w and $O(N_{in}M{\left(N_{T}L\right)}^{2})$ operations to tackle (39) with $N_{in}$ denoting the number of (inner) iterations of the proposed fractional programming.

For the computational complexity of both transceiver joint optimization algorithms under similarity constraint, the discussion is similar to algorithms under constant-envelope constraint.

## 5. Simulation Results

Consider a side-looking MIMO radar with 4 transmit antennas and 4 receiver antennas, the direction of the radar’s linear array is parallel to the radar’s moving direction. The simulation parameters are are organized into three groups: target parameters, jamming parameters and clutter parameters in Table 1.

Figure 2 shows the space-time cross-ambiguity of different optimizations. The space-time cross-ambiguity is defined by:(57)$$P_{w,S}={\left|w^{H}{\tilde{S}}^{T}v_{t}\right|}^{2}={\left|w^{H}\left(D\left(f_{d}\right)⊗I_{L}⊗A\left(\theta_{t}\right)\right)\hat{s}\right|}^{2}.$$

Compare the detection results of the joint optimization with fixed pulse-to-pulse waveform and pulse-to-pulse waveform variations, both are under energy constraint. From the space-time cross-ambiguity results, we observe that the mainlobe of both the optimization is at the moving target’s location: the normalized Doppler frequency is 0.3 (the corresponding speed is 45 m/s) and the normalized spatial frequency is 0 (the corresponding azimuth is $0^{\circ }$). It can be seen from the comparison result that the optimization with pulse-to-pulse variations has a better clutter and jamming suppression performance, so that the target detection result is more accurate.

For the sake of completeness, we also conduct simulations of a larger MIMO radar configuration with 6 transmit antennas and 6 receiver antennas and other parameters are the same as the simulations above. Figure 2c shows the Space-time cross-ambiguity of this radar system. The result shows that the algorithm is also valid in larger MIMO radar configuration and the concentration of the main-lobe is better than the 4×4 radar configuration, the output SINR is also increased as their are more antennas. Based on the author’s current research projects and the next research plan, subsequent simulations are still based on MIMO radar with 4 transmit antennas and 4 receiver antennas.

Then we discuss the convergence of the two optimizations. Compare the relationship between output SINR and iteration number of joint optimization with fixed pulse-to-pulse waveform and pulse-to-pulse waveform variations, also use the optimization under energy constraint as a benchmark. The comparing result is shown in Figure 3.

From Figure 3, the output SINR of both two methods contract to maximum: the output SINR of joint optimization with fixed pulse-to-pulse waveform contracts to 22.87 dB after 12 iterations and the output SINR of joint optimization with pulse-to-pulse waveform variations contracts to 23.58 dB after 16 iterations. The computation complexity of the generalized approach is higher because the waveform is varying between pulses and the iteration number is increasing. Although the method with pulse-to-pulse waveform variations needs more iterations and the contract speed is slower but this method obviously increases the output SINR, which means the method achieves a more accurate detection result.

The advantage of joint optimization with pulse-to-pulse waveform variations also reflected in the minimum detectable velocity (MDV), which is defined as the velocity closest to that of the main-lobe clutter at which acceptable SINR loss is achieved. Compare the MDV achieved by joint optimization results of fixed pulse-to-pulse waveform and pulse-to-pulse waveform variations. Moreover, linear frequency modulation (LFM)waveform is plotted as benchmark. Here, the LFM waveform means a group of scaled version of LFM waveform. The receive filters of the LFM waveforms are designed by (12). The comparison of MDV is reflected by the SINR loss of the three methods respectively, the comparison result is shown in Figure 4.

It can be seen from the comparison result that all the methods have strong suppressing effect when the Doppler frequency is close to 0, so all of them achieve relatively well clutter suppression performances. Under the allowed SINR loss, the narrower the notch, the lower the MDV, the better the detection effect. A method with low MDV means it can detect target with very low velocity. Figure 4 focuses on the low Doppler frequency part, as it is of greater interest of the applications. A detecting method has better performance at low Doppler frequency means it has better performance in detecting low-velocity target. But at high Doppler frequency, the performance of LFM waveforms is similar to our algorithm. The result in Figure 4 shows that the joint optimization with pulse-to-pulse waveform variations has the lowest MDV, it proves that this method has the best detection performance.

Then we use a set of numerical simulations to observe the difference in performances of the optimizations after adding the constant-envelope constraint and similarity constraint.

Firstly, consider the constant-envelope constraint. Also consider a side-looking MIMO radar with 4 transmit antennas and 4 receiver antennas, the direction of the radar’s liner array is parallel to the radar’s moving direction. The simulation parameters of the system are shown in Table 1, set the randomization parameters: r = 100.

Firstly, the relationship between the output SINR and the number of iterations of the joint optimization with pulse-to-pulse waveform variations by methods which Section 4.2.1 and Section 4.2.2 sections are analyzed. (For the sake of simplicity, let method 1 refers to the optimization method based on relaxation and randomization; method 2 refers to method based on the fractional programming and the power-like iterative), the threshold ε for determining whether the output SINR has converged is 1 × 10${}^{-3}$ and the result is shown in Figure 5:

From the figure, the output SINR of the three algorithms is convergent. The energy-constrained joint optimization algorithm converges to 23.58 dB after 28 iterations and the constant-envelope constraint method 1 converges to 23.44 dB after 24 iterations; method 2 converges to 23.46 dB after 21 iterations. It can be seen that after the constant-envelope constraint is added, the resulting output SINR reaches convergence and the level does not decrease significantly.

The relationship between the output SINR and the Doppler frequency of Method 1 and Method 2 under constant-envelope constraint is compared. Based on the optimization method under energy constraint, the comparison results are shown in Figure 6:

From the comparison results in the Figure 6, all of the three methods have strong suppression effect when the Doppler frequency is close to zero and the notches are relatively narrow, indicating that the purpose of suppressing the clutter is well achieved. This also shows that the performance of the joint optimization method will not be significantly reduced after the constant-envelope constraint is added.

Then we consider the similarity constraint. Also consider a side-looking MIMO radar with 4 transmit antennas and 4 receiver antennas, the direction of the radar’s liner array is parallel to the radar’s moving direction. The simulation parameters of the system are shown in Table 1.

Firstly analyse under the similarity constraint when the similarity parameters are different, how the relationship between the output SINR and the number of iterations of the algorithm changes (for the sake of simplicity, the optimization method based on relaxation and rank 1 decomposition is the method 1, based on fractional programming and the SWORD method method are the methods 2), the threshold ε for determining whether the output SINR has converged is 1 × 10${}^{-3}$ and the similarity parameters δ are taken $1e_{t},0.5e_{t},0.1e_{t}$ separately. The method 1 of the similarity constraint is taken as an example. The relationship between the output SINR and the number of iterations processed by the method 1 under the similarity constraint is shown in Figure 7:

It can be seen from the results in the figure that when the similarity parameter increases, the output SINR performance of the algorithm will increase correspondingly but the convergence performance of the algorithm will decrease accordingly. The similarity parameters are respectively converge to 23.58 dB, 22.64 dB and 19.63 dB after 24, 17 and 12 iterations.

The relationship between the output SINR and the Doppler frequency of Method 1 and Method 2 under constant mode constraint is compared. Based on the optimization method under energy constraint, the comparison results are shown in Figure 8:

From the comparison results in the figure, the coincidence of the three curves is very high and all of the three have strong suppression effects when the Doppler frequency is close to zero. The notches of the three methods are relatively narrow, which means that the three methods achieve the purpose of suppressing clutter better. This also shows that the performance of the joint optimization method will not be significantly reduced after the constant-envelope constraint is added.

## 6. Conclusions

The MIMO GMTI based on non-orthogonal waveform improves the detection performance in specific tasks, especially the detection of low-velocity moving targets. In this paper, we conduct the joint optimization under maximum output SINR criterion, improve the target detection performance by cyclic joint optimization of the receiver filter and transmit waveform. We also generalize the optimization to MIMO radar system with pulse-to-pulse waveform variations, discuss the optimizations under energy constraint, constant-envelope constraint and similarity constraint. Compared to optimization with fixed pulse-to-pulse waveform, we prove that the optimization with pulse-to-pulse variations obviously improves the output SINR and the optimizations under all the three constraints gets relatively good jamming and clutter suppressing performances. Also, cyclic optimization is used to achieve the transceiver joint design, so as to get a better suppressing performance of clutter and noise. Given that the joint optimization with pulse-to-pulse variations increases the detection performance, it also increase the algorithm’s computation complexity. To make the optimization method more practical, more efficient and fast algorithm should be studied. 

## Figures and Tables

**Figure 1 sensors-19-05575-f001:**
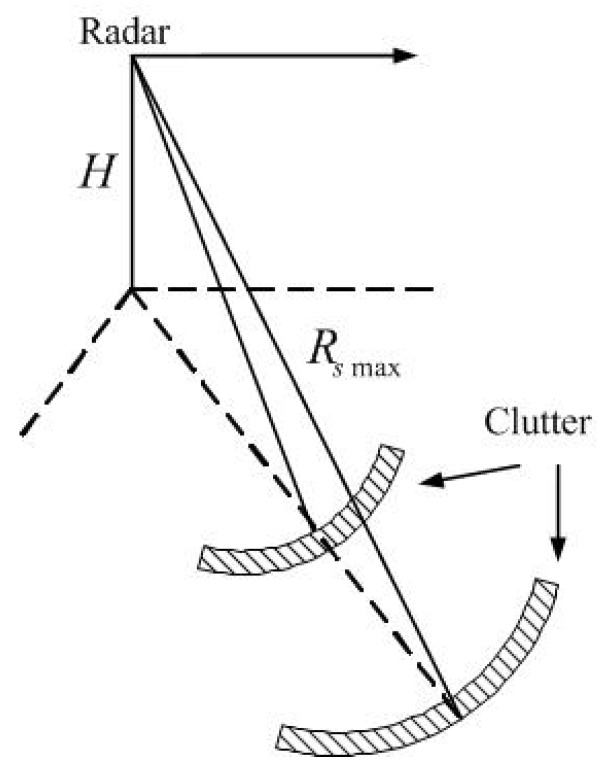
Schematic diagram of clutter unit division.

**Figure 2 sensors-19-05575-f002:**
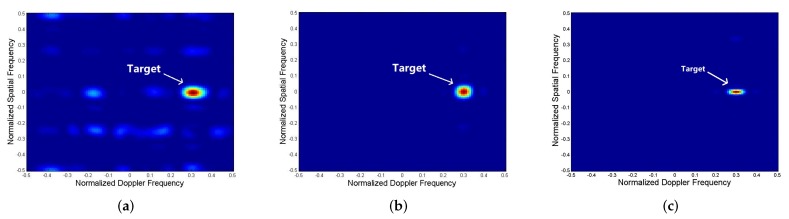
Space-time cross-ambiguity of (**a**) Joint optimization with fixed pulse-to-pulse waveform under energy constraint, (**b**) Joint optimization with pulse-to-pulse waveform variations, (**c**) Joint optimization with pulse-to-pulse waveform variations with 6×6 radar configuration.

**Figure 3 sensors-19-05575-f003:**
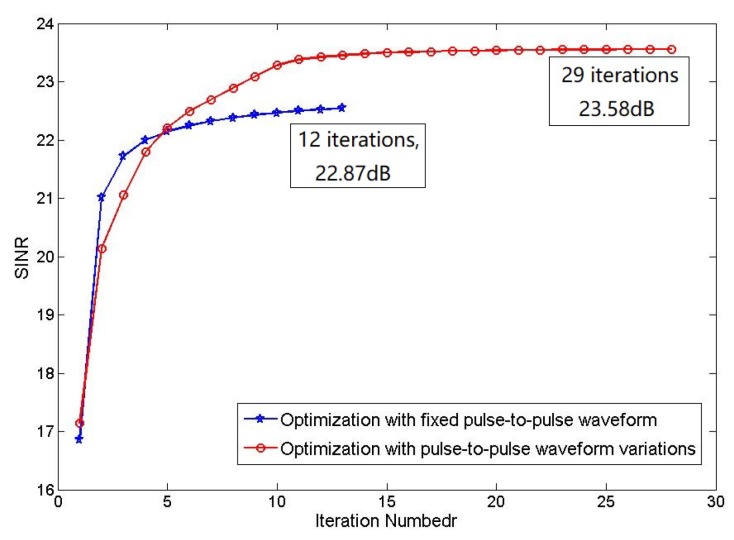
Comparison of the relationship between output signal interference and noise ratio (SINR) and iteration number.

**Figure 4 sensors-19-05575-f004:**
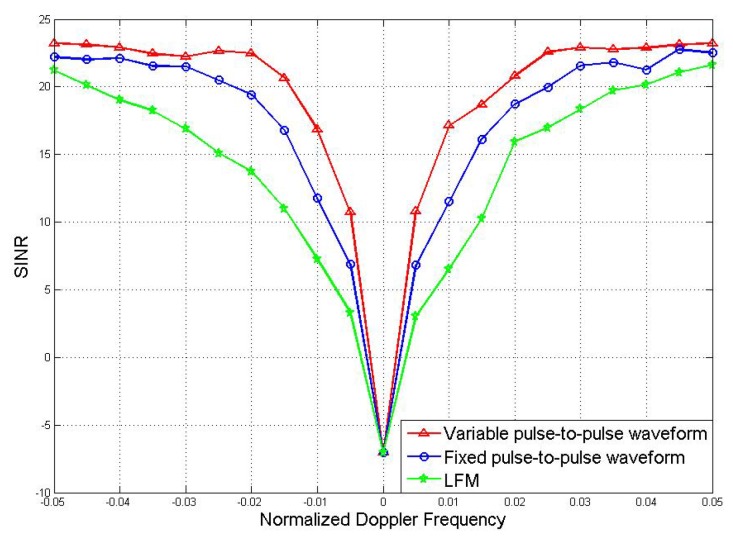
Comparison of minimum detectable velocity (MDV).

**Figure 5 sensors-19-05575-f005:**
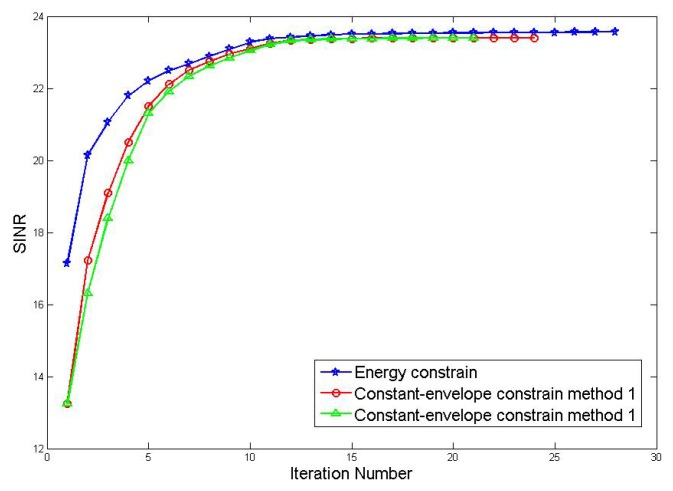
Comparison of iteration number.

**Figure 6 sensors-19-05575-f006:**
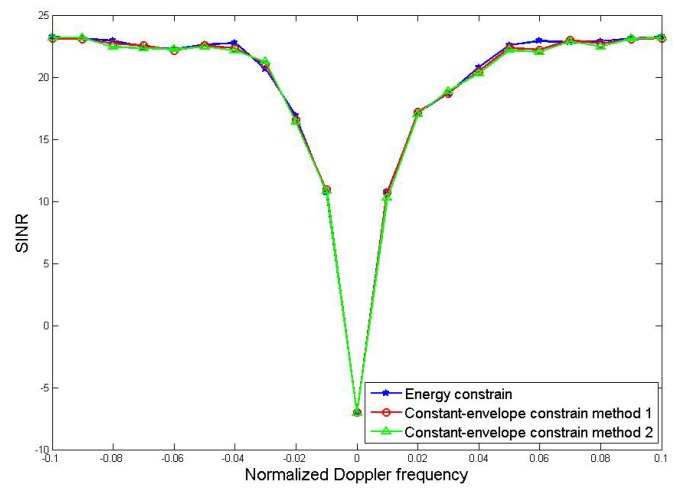
Comparison of MDV.

**Figure 7 sensors-19-05575-f007:**
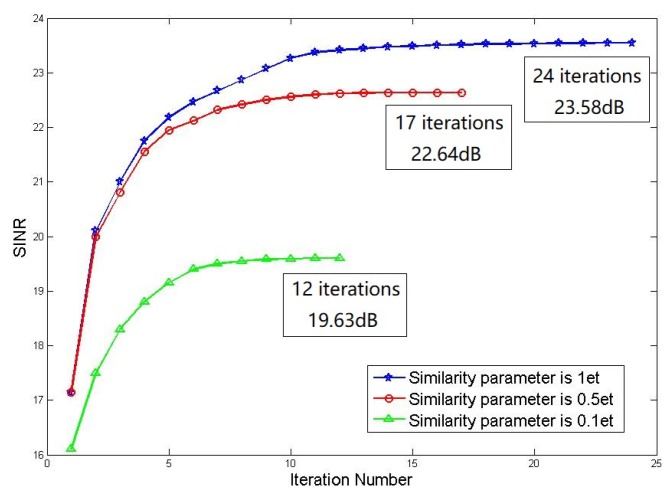
Comparison of iteration number.

**Figure 8 sensors-19-05575-f008:**
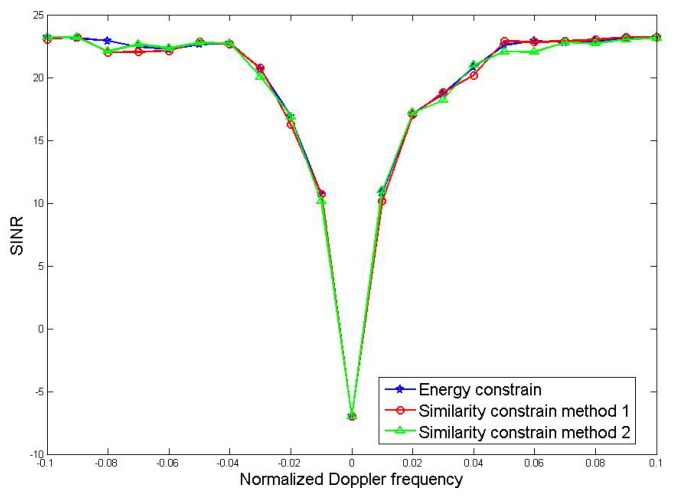
Comparison of MDV.

**Table 1 sensors-19-05575-t001:** Simulation Parameters.

Parameters	Symbol	Value
Radar system parameters		
Transmitter number	$N_{T}$	4
Receiver number	$N_{R}$	4
CPI	*M*	16
Transmit energy	$e_{t}$	1
Carrier frequency	$f_{0}$	1 GHz
Inter-element spacing of receivers	$d_{r}$	0.15 m
Wavelength	λ	0.3 m
PRF	$f_{r}$	1000 Hz
Platform velocity	$V_{a}$	150 m/s
Platform hight	*H*	9000 m
Code length	*L*	8
Inter-element spacing of transmitters	$d_{t}$	0.6 m
Cyclic threshold value	ε	1 × 10${}^{-3}$
Similarity constraint value	δ	$e_{t}$
Target parameters		
Target range	$R_{t}$	12,728 m
Target velocity	$v_{t}$	45 m/s
Target azimuth	$\theta_{t}$	0${}^{\circ }$
Jamming parameters		
Jamming direction	$\theta_{j}$	$30^{\circ }$
Jammer to noise ratio	JNR	35 dB
Clutter parameters		
Clutter patch number	$N_{c}$	361
Clutter patch variance	$\sigma_{c,l,k}^{2}$	1
Nearest clutter patch number	*K*	1

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
