# Peer review of "Joint Optimization of Transmit Waveform and Receive Filter with Pulse-to-Pulse Waveform Variations for MIMO GMTI"

_sensors, 2019, doi:10.3390/s19245575_

Round 1

Reviewer 1 Report

Dear authors,

MIMO SAR is a topic of high interest with regard to future SAR missions. In this context, your investigations concerning the application of MIMO SAR in GMTI are a valuable contribution to the scientific discussion in this area of research. Your approach, in which way you jointly optimize the two major parameters occuring in the investigated use case - the pulse waveform and the receive filter - is remarkable.

The numerous cited references indicate in their number and choice your good overview on the scientific discussion in your area of research.

Unfortunately, there is also one point of critique: With regard to English usage and style, I see your paper in need for a thorough revision. I just can discuss few repeatedly occuring mistakes, here. The support by an experienced English speaker may be of help for you.

You frequently use the word "could" which relativizes the statement of your sentences. In some cases the usage of "can" is more adequate. In most cases, you neither need "could" nor "can" as both words imply a possibility where you actually observed a real fact. E.g. in the first sentence of your conclusions (page 16, line 372/373), you wrote: "The MIIMO GMTI based on non-orthogonal waveform could improve the detection performance in specific tasks, [...]" The usage of past tense implies, that your observations were valid only in the past but not any more. I see no need for this restriction: You thoroughly investigated the performance of your approach in the experimental part of your research and the results show that the approach actually improves the detection performance. So just use "improves" without any "could" or "can".

A frequently occuring typo is the omission of the trailing character "t" in the noun "constraint" (maybe caused by a confusion with the spelling of the associated verb "to constrain").

You also may take a closer look on the usage of infinitive and present paticiple ("-ing"). You regularly use the first one where the latter one is appropriate. e.g. correct page 1, line 5: "performance in supress clutter and noise" -> "performance in suppressing clutter and noise". Page 2, line 47: "Follow the ideas above" -> "Following the ideas above"

A rather minor remark: In the chapter "Simulation results" you continously present measurement results with 4 positions after decimal point. Do you really expect that all of these digits are significant? Maybe, you can omit the last two digits.

Please spend the time and effort in thoroughly revising your paper as your approach is worth to be published and to be considered in the scientific discussion.

Author Response

Thank you for your valuable suggestions for this article. For the response of your suggestions, please see the attachment.

Reviewer 2 Report

Overall, the present work is a nice study on joint design of transmit-receive filter with MIMO radar and GMTI applications. However, before publication, it is my opinion that the following major comments should be addressed by the authors:

1) Abstract – “Multi-input multi-output (MIMO) is usually defined as a radar system which in the transmit time and receive time, space and transform domain can be separated into multiple independent
signals.” ->

“Multi-input multi-output (MIMO) radar is usually defined as a radar system in which the transmit time and receive time, space and transform domain can be separated into multiple independent
signals.”

2) The following work on transmit-receive filter design (with covert communications focus) has been missed by the authors:

"Intrapulse radar-embedded communications via multiobjective optimization." IEEE Transactions on Aerospace and Electronic Systems 51.4 (2015): 2960-2974.

3) Please add an organization paragraph at the end of Sec. I.

4) For the sake of better readability, it would be nicer moving the notation paragraph at the end of Sec. I.

5) When discussing the advantages of MIMO radar in Sec. I, it would be also nice mentioning the improved localization performance attainable thanks to the improved spatial diversity, e.g.

"Target localization accuracy gain in MIMO radar-based systems." IEEE Transactions on Information Theory 56.6 (2010): 2783-2803.

"Performance analysis of time-reversal MUSIC." IEEE Transactions on Signal Processing 63.10 (2015): 2650-2662.

6) Sec. 2.1 – A few more details on how Eq. (1) is obtained should be given by the authors.

7) For the considered optimization approaches it would be very important to provide a graphical block scheme depicting the workflow of each of them.

8) Similarly, the considered optimization approaches should be also compared in terms of computational complexity.

9) The readability of all the figures should be improved in terms of graphical rendering.

10) For the sake of completeness, it would be also useful showing performance for a large MIMO radar configuration (say 6x6).

11) Please double-check the whole bibliography, e.g. the year and the publication venue of [33] seems wrong.

12) Conclusions should be enriched with what the authors consider to be further avenues of research.

13) Finally, the paper should be carefully proof-read from typos, e.g.:
Sec. I – “to be possible[7]–[9].Its” -> “to be possible [7]–[9]. Its”
Sec I – “has also been significantly improved[3].” -> “has also been significantly improved [3].”
Sec. I – “minimum detectable velocity (MDV)[10]–[13].” -> “minimum detectable velocity (MDV) [10]–[13].”
Sec. I – “(MI) criterion[17]–[20],” -> “(MI) criterion [17]–[20],”
Sec. I – “and ambiguity function[22,23]” -> “and ambiguity function [22,23]”.
Sec. II.4 – “addictive white Gaussian noise” -> “additive white Gaussian noise”
Title of Sec. IV.1 – “Joint optimization under energy constrain” -> “Joint optimization under energy constraint” (I have also found other instances throughout the manuscript).
Sec. V – “the radar’s liner array” -> “the radar’s linear array”.

Author Response

(The authors gave the same response as above.)

Round 2

Reviewer 2 Report

Overall, the present work is a nice study on joint design of transmit-receive filter with MIMO radar and GMTI applications.

Additionally, the authors have satisfactorily addressed my previous comments and modified their manuscript accordingly. Hence, I am glad to recommend the present work for publication.

This manuscript is a resubmission of an earlier submission. The following is a list of the peer review reports and author responses from that submission.

Round 1

Reviewer 1 Report

The paper is interesting and the numerical results promising. Nevertheless before a possible publication the following comments should be carefulllyy addressed.

A significant retyping process is required to improve English exposition.

Rephrase the sentence ''In this paper, we chose maximum output signal 6 interference and noise ratio (SINR) criteria, joint design the transmit waveform and receive filters 7 by cyclic optimization''

It could be worth to quote and discuss relevant works that have been published in the context of transmit signal and receive filter design, e.g.

"Knowledge-Aided (Potentially Cognitive) Transmit Signal and Receive Filter Design in Signal-Dependent Clutter," IEEE Transactions on Aerospace and Electronic Systems, 2013.

"Optimization of the receive filter and transmit sequence for active sensing" IEEE Transactions on Signal Processing 2012. 

"Waveform design in signal-dependent interference and application to target recognition with multiple transmissions" IET 2009. 

"Optimizing Radar Waveform and Doppler Filter Bank via Generalized Fractional Programming," IEEE Journal of Selected Topics in Signal Processing, 2015.

‘’Knowledge-based design of space–time transmit code and receive filter for a multiple-input–multiple-output radar in signal-dependent interference’’, IET 2015

"Robust Waveform and Filter Bank Design of Polarimetric Radar," IEEE Transactions on Aerospace and Electronic Systems, vol. 53, no. 1, pp. 370-384, Feb. 2017.

"Cognitive design of the receive filter and transmitted phase code in reverberating environment", IET 2012

Moreover, the authors should also highlight the importance of MIMO system to provide more flexibility in the beampattern design e.g.,

"MIMO radar with colocated antennas" IEEE Signal Processing Magazine, 2007. 

"Waveform synthesis for diversity-based transmit beampattern design" IEEE Trans. Signal Processing 2008. 

"MIMO Radar Beampattern Design Via PSL/ISL Optimization," in IEEE Transactions on Signal Processing, 2016.

It is not clear if the clutter is ambiguous or not. Also, why the clutter return looks like different from the target one?

The clutter Doppler should present some randomness.

Rephrase the sentence ''During the simulation process''

As to Problem (25) please refer also to

"Cognitive design of the receive filter and transmitted phase code in reverberating environment", IET 2012

Moreover, you might also solve it with convergence guarantee resorting to

"A New Sequential Optimization Procedure and Its Applications to Resource Allocation for Wireless Systems," IEEE Transactions on Signal Processing, 2018.

In particular you may optimize one phase at time as well as the receive filter in either a cyclic way or resorting to the MBI.

As to Problem (37) please refer also to

"Knowledge-Aided (Potentially Cognitive) Transmit Signal and Receive Filter Design in Signal-Dependent Clutter," IEEE Transactions on Aerospace and Electronic Systems, 2013.

The numerical results should be better illustrated and commented as well as some additional case studies should be included. Moreover, improve the captions and specify all the parameters, e.g., the similarity constraint value.

Author Response

Thank you for your comments concerning to our manuscript entitled “Joint Optimization of Transmit Waveform and Receive Filter with Pulse-to-Pulse Waveform Variations for MIMO GMTI” (ID: sensors-560769). Those comments are very helpful for revising and improve our paper. The important guiding is significance to our research. We have studied the comments carefully and have made corrections which we hope to meet approval. Revised portions are marked in red in the paper. Please see the attachment for response.

Reviewer 2 Report

1. In (1), some symbols are left undefined, e.g., $\mathbf{I}_L$, the Kronecker product.  Besides, the dimension of waveform $\mathbf{S}$, $\mathbf{T}_m$, $\mathbf{t}_m$,… are all undefined, makes the signal model unclear to the readers.

2. In (8), please use small case when writing $k_B$.

3. This paper models the signal as well as clutter in optimization of waveforms under energy constraint, constant-envelop constraint and similarity constraint.  It is not convincing that the constant-envelop constraint results in better clutter suppression performance.  Further comments should be provided.

4. In Figure 4, the reason why LFM waveform behaves the worst?  This is not convincing because it is not clear how the LFM waveform is used.  See the following paper the references therein for different viewpoint.

“A range ambiguity resolution approach for high-resolution and wide-swath SAR imaging using frequency diversity array,” IEEE Journal of Selected Topics in Signal Processing, March 2017.

5. Anyway, joint waveform design and receive filtering is an interesting topic in MIMO radar. Different waveforms can be applied in both slow-time and fast-time, resulting in sophisticated waveforms.

Author Response

(The authors gave the same response as above.)

Reviewer 3 Report

The paper conduct the joint optimization under maximum output SINR criteria, improve the target detecting performance by cyclic joint optimize the receiver filter and transmit waveform. Generalize the optimization to MIMO radar system with pulse-to-pulse waveform variations, discuss the optimizations under energy constrain, constant-envelope constrain and similarity constrain. Compare to optimization with fixed pulse-to-pulse waveform, proved that optimizations with pulse-to-pulse variations could obviously improve the output SINR and the optimizations under all the three constrains could get a relatively good jamming and clutter suppressing performance. Also, cyclic optimization is used to achieve the transceive joint design, so as to get a better suppressing performance of clutter and noise. The simulation results presented in the paper are convincing. This paper could be published in Sensors.

Author Response

Thank you for your comments concerning to our manuscript entitled “Joint Optimization of Transmit Waveform and Receive Filter with Pulse-to-Pulse Waveform Variations for MIMO GMTI” (ID: sensors-560769). Those comments are very helpful for revising and improve our paper. The important guiding is significance to our research. We have studied the comments carefully and have made corrections which we hope to meet approval. Revised portions are marked in red in the paper. 

Reviewer 4 Report

Great body of work! But, this paper could benefit from a better discussion of the figures as compared to the text.

Author Response

Thank you for your comments concerning to our manuscript entitled “Joint Optimization of Transmit Waveform and Receive Filter with Pulse-to-Pulse Waveform Variations for MIMO GMTI” (ID: sensors-560769). Those comments are very helpful for revising and improve our paper. The important guiding is significance to our research. We have studied the comments carefully and have made corrections which we hope to meet approval. Revised portions are marked in red in the paper. 

Thank you for your valuable suggestions, the discussion part of the paper has been modified, and we will continue to explore this research direction.

Reviewer 5 Report

This paper considers joint optimization of transmit waveforms and receive filters for MIMO radar, and extends previous work to optimize waveforms on a pulse-by-pulse basis. I have 2 main comments on this manuscript.

This paper is poorly written. There are numerous grammar mistakes and typos throughout the text. The authors have not demonstrated the utility of their approach. In Section 4, they correctly state that their approach has higher computational complexity but may achieve better performance when compared to fixed pulse-to-pulse waveforms. What is the added complexity? What is the enhanced performance? The results from Figure 3 show an improvement of 0.7 dB, which is not significant.

Author Response

Thank you for your comments concerning to our manuscript entitled “Joint Optimization of Transmit Waveform and Receive Filter with Pulse-to-Pulse Waveform Variations for MIMO GMTI” (ID: sensors-560769). Those comments are very helpful for revising and improve our paper. The important guiding is significance to our research. We have studied the comments carefully and have made corrections which we hope to meet approval. Revised portions are marked in red in the paper. Please see the attachment for responses.

Round 2

Reviewer 1 Report

The authors have significantly improved their work after this revision round. Nevertheless additional care should be given to the English exposition. Moreover, the author didn't properly reply to the previous comments 5. In particular, they should clearly highlight that 

As to Problem (25) please note that you might also solve it with convergence guarantee resorting to
"A New Sequential Optimization Procedure and Its Applications to Resource Allocation for Wireless Systems," IEEE Transactions on Signal Processing, 2018.
In particular you may optimize one phase at time as well as the receive filter in either a cyclic way or resorting to the MBI.

Please, it could be worth to clearly highlight this fact in the revised version of the paper

Author Response

Thank you for your suggestions, the methods of the paper has been highlighted in our manuscript, and the modified place has been marked red. 

Reviewer 5 Report

In the latest version of the manuscript, the authors have made some changes.  However, the main comments from my first round review are still valid.  There are still numerous grammar and spelling mistakes throughout the text.  In my opinion, the improvements in SINR and MDV are not significant enough to warrant acceptance.